# LBI-FL: Low-Bit Integerized Federated Learning with Temporally Dynamic Bit-Width Allocation

**Li Ding** [* 1]  **Hao Zhang** [* 1]  **Wenrui Dai** [2]  **Chenglin Li** [1]  **Weijia Lu** [3]
**Zhifei Yang** [3]  **Xiaodong Zhang** [3]  **Xiaofeng Ma** [3]  **Junni Zou** [2]  **Hongkai Xiong** [1]

## Abstract

Federated learning (FL) is greatly challenged by the communication bottleneck and computation limitation on clients. Existing methods based on quantization for FL cannot simultaneously reduce the uplink and downlink communication cost and mitigate the computation burden on clients. To address this problem, in this paper, we propose the first low-bit integerized federated learning (LBI-FL) framework that quantizes the weights, activations, and gradients to lower than INT8 precision to evidently reduce the communication and computational costs. Specifically, we achieve dynamical temporal bit-width allocation for weights, activations, and gradients along the training trajectory via reinforcement learning. An agent is trained to determine bit-width allocation by comprehensively considering the states like current bit-width, training stage, and quantization loss as the state. The agent efficiently trained on small-scale datasets can be well generalized to train varying network architectures on non-independent and identically distributed datasets. Furthermore, we demonstrated in theory that federated learning with gradient quantization achieves an equivalent convergence rate to FedAvg. The proposed LBI-FL can reduce the communication costs by 8 times compared to full-precision FL. Extensive experiments show that the proposed LBI-FL achieves a reduction of more than 50% BitOPs per client on average for FL with less than 2% accuracy loss compared to low-bit training with INT8 precision.

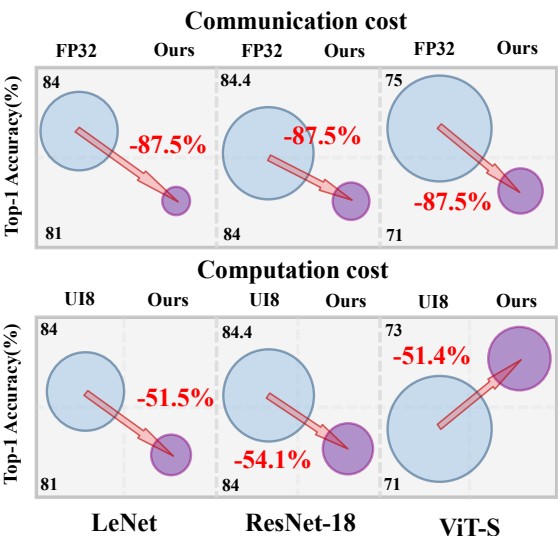

Figure 1: Image classification results on CIFAR-10 with the low-bit integerized federated learning framework (LBI-FL). The proposed method can simultaneously reduce both the communication cost and computational cost. The areas of bubbles are positively correlated with the communication cost or the BitOPs for training the model.

## 1. Introduction

Federated learning (FL) is a distributed machine learning framework designed to enable model training without the local data exchange. In FL, multiple clients independently train a model on their local data and periodically send the updated model parameters (or model updates) to a central server for aggregation. This paradigm can address the data privacy and security concerns while effectively leveraging the computational resources distributed over local clients.

However, a significant challenge in FL is the communication overhead, as it necessitates a frequent transmission of models or updates between local clients and the central server. To address this, a substantial body of literature has focused on compressing the model updates through techniques such as quantization and sparsification. Compression-based federated algorithms such as QSGD (Alistarh et al., 2017)

---
*Equal contribution  [1]Department of Electronic Engineering, Shanghai Jiao Tong University, Shanghai, China. [2]Department of Computer Science and Engineering, Shanghai Jiao Tong University, Shanghai, China. [3]United Automotive Electronic Systems, Shanghai, China. Correspondence to: Wenrui Dai <daiwenrui@sjtu.edu.cn>, Chenglin Li <lcl1985@sjtu.edu.cn>.

and SignSGD (Bernstein et al., 2018) reduce the amount of model updates to be uploaded to the server by quantizing the gradient values into lower-precision integers. Though these methods can mitigate the uplink communication cost, they still require downloading a full-precision model. Exploring lower bit-width training in both the uplink and downlink communication becomes promising to further reduce the communication overhead for FL systems.

Network quantization (Hubara et al., 2017; Jacob et al., 2018), as a mainstream strategy to achieve network compression, has been extensively studied for network inference and training, since it enjoys the merit of reducing computation burden without modifying the model architecture. Different from inference quantization (Jacob et al., 2018; Banner et al., 2019; Choi et al., 2018; Jung et al., 2019) that quantizes the weights and activations of networks to accelerate the inference process for deployment, low-bit training (training quantization) simultaneously projects gradients, weights, and activations into low bit-width during training. Thus, low-bit training can reduce the uplink and downlink communication overhead for federated learning using low-bit representation of weights for training. Moreover, it can decrease the computation load for each client by reducing the computational cost of backward computation (backpropagation) that is nearly twice the cost for forward computation (Zhao et al., 2021).

Recent attempts on low-bit training, including directive sensitive gradient clipping (Zhu et al., 2020) and vectorized gradient quantization (Zhao et al., 2021), can alleviate or even eliminate the accuracy loss of networks trained under the INT8 precision. However, these methods could collapse when the precision is further decreased to INT4. Considering that neural networks vary in sensitivity to quantization at different training stages, it is reasonable to assign different bit-widths for quantization at different stage of network training to minimize the impact of quantization error. In this paper, we focus on a mixed precision of INT4, INT6, and INT8 for dynamic temporal bit-width assignment to reduce both communication and computation costs for FL. Zhang et al. (2020) use a predetermined threshold to decide layer-wise switching of bit-widths for gradients with fixed bit-widths for weights and activations. Nevertheless, as shown in Figure 2, the quantization loss varies for different networks during the training process and a predetermined threshold cannot achieve consistent compression ratio for varying networks or datasets. There is a lack of a mixed-precision bit-width allocation that can dynamically fit the varying training processes for different networks on non-independent and identically distributed training data.

In this paper, we propose the first low-bit integerized federal learning (LBI-FL) framework that allows an average precision below INT8 to further reduce the communication and

computation costs at a tolerable level of performance loss as shown in Figure 1. We leverage reinforcement learning (RL) to dynamically determine the bit-widths for weights, activations, and gradients by comprehensively considering the quantization loss along the training trajectory. An agent is efficiently trained on a small local dataset and generalized to large-scale models and complicated datasets. Our main contributions are summarized as follows.

- To our best knowledge, this is the first successful attempt to achieve low-bit training FL that evidently reduces the communication overhead and computation cost compared to full-precision and INT8 training.

- We propose a novel reinforcement learning method for temporally dynamic bit-width allocation for weights, activations and gradients along the training trajectory to achieve an average bit-width below INT8.

- We demonstrate in theory that federated learning with gradient quantization achieves equivalent convergence rate to the standard FedAvg algorithm (McMahan et al., 2017) with sufficiently large number of communication rounds and further empirically verify the convergence rate.

- Comprehensive experiments on low-bit integerized federated learning with a wide range of network models including ResNets and ViTs validate the effectiveness of the proposed method.

## 2. Related Work

**Inference quantization.** According to the stage of quantization, model quantization can be divided into two categories: inference quantization and training quantization. Inference quantization quantizes the weights or activations of the trained neural networks from full precision to low-bit data format representations before deploying the models. Typical works (Hubara et al., 2016; Rastegari et al., 2016; Liu et al., 2023) utilize the binary or ternary bit-widths for weights and activations to speed up the inference. Dong et al. (2019; 2020) achieve a good compression performance through the mixed-precision quantization.

**Training quantization.** Training quantization stands for quantizing simultaneously the gradients, weights, and activations to accelerate the training process. There are two main types of methods in existing studies of low-bit training. The first is to use low-bit floating point for training (Wang et al., 2018; Mellempudi et al., 2019; Cambier et al., 2020). This method can achieve almost lossless accuracy with the full-precision model, but harms the acceleration performance compared to integer quantization. The second involves quantizing the model with integers. Some methods quantize the entire network (Zhou et al., 2016; Yang et al., 2020a)

which yields a higher compression but harms the model performance. Some other methods only quantize the convolutional layers in the model (Zhu et al., 2020; Zhao et al., 2021) which can achieve a better model performance. The low-bit training framework in this paper is closest to these methods, but the goal is to further reduce the bit-width on the existing full INT8 basis for a more efficient FL scheme.

**Federated learning.** Federated learning, as a rapidly evolving application of distributed learning in large-scale client networks, has garnered significant research interest. This surge of attention has led to a substantial body of work exploring the intersection of FL with various domains, including robustness (Ghosh et al., 2019), fairness (Wei & Huang, 2024), federated reinforcement learning (Yue et al., 2024), and federated optimization (Reddi et al., 2021). For a more detailed comparison, we defer to the comprehensive survey papers (Li et al., 2020; Kairouz et al., 2021).

**Efficient FL.** Given the communication bottlenecks and computational resource constraints on the client side in FL, introducing pruning (Prakash et al., 2022; Meinhardt et al., 2024) or quantization techniques is a natural progression, and we mainly introduce the quantization methods here. To enhance communication efficiency, various methods have adopted quantized SGD (Alistarh et al., 2017; Bernstein et al., 2018; Mishchenko et al., 2022), where model updates (i.e., the sum of gradients) are directly quantized and compressed. Some methods (Li & Li, 2023; Bernstein et al., 2018) also incorporate error feedback to mitigate quantization errors. However, these quantized model updates primarily reduce the uploaded traffic without accelerating training or alleviating downloaded traffic. To further expedite the local training and inference, certain strategies (Chen et al., 2024) explore the use of quantized neural networks within FL. Nevertheless, these methods often treat the problem as an optimization task, focusing on determining the optimal quantization bit-width. While theoretically performant, their empirical results exhibit a slight decrease in the model effectiveness. Moreover, some methods improve the FL efficiency by optimizing energy consumption. Yang et al. (2020b) derive the time energy consumption models and Marnissi et al. (2024) propose an optimization framework to minimize the total energy consumption.

## 3. Preliminaries

**Quantization.** Integer quantization maps a floating-point value to a fixed-point number. Uniform quantization is popular in network quantization. It is classified into two kinds, *i.e.*, symmetric quantization and asymmetric quantization, based on whether the mapping is identical for the zero point.

Asymmetric quantization first substracts the zero-point $z$ from the clipped data $x'_f = \text{clamp}(x_f, m, M)$ and then

multiplies the data by the scaling factor $s$. Here, $M$ and $m$ are the maximal and minimal clipping values for $x_f$, respectively. The scaled data is fed into the rounding function,

$$x_q = \text{round}\left(\frac{1}{s} \cdot (x'_f - m)\right) = \text{round}\left(\frac{1}{s} \cdot x_f + z\right), \quad (1)$$

where the scaling factor $s = (M - m)/(2^N - 1)$ given the bit-width $N$ for quantization. De-quantization is realized by $\hat{x}_f = (x_q - z) \cdot s$. Different from asymmetric quantization, symmetric quantization obtains zeros for the zero point.

$$x_q = \text{round}\left(\frac{1}{s} \cdot x'_f\right), \ \hat{x}_f = x_q \cdot s, \ s = \frac{\max(|x_f|)}{2^{N-1} - 1}. \quad (2)$$

Nearest rounding is usually used for the weight and activation quantization, while stochastic rounding is adopted (Gupta et al., 2015) for the gradient quantization, since its loss has much more impact on model accuracy,

$$\text{round}(x) = \begin{cases} \lfloor x \rfloor, & \text{w.p.} \quad 1 - (x - \lfloor x \rfloor) \\ \lfloor x \rfloor + 1, & \text{w.p.} \quad x - \lfloor x \rfloor \end{cases} \quad (3)$$

where $\lfloor x \rfloor$ returns the largest integer not greater than $x$.

**Deep Q-Network.** Reinforcement learning optimizes the accumulative rewards interacting with the environment. The interaction between the agent and environment can be modeled as a Markov decision process (MDP) represented by a five-tuple $(s, a, t, r, \gamma)$ of state $s$, action $a$, transition function $t$, reward function $r$, and discounting factor $\gamma$.

The Q-function $q^\pi(s, a)$ is defined as the expectation reward of taking specific action on the current state. Deep Q-network (DQN) is a classic value-based algorithm using a neural network $q_\theta(s, a)$ to approximate this Q-function $q^\pi(s, a)$. DQN consists two important components, *i.e.*, target network and experience replay (Mnih et al., 2015). The target network stabilizes the training results and experience replay diversifies the batch data. In this paper, we employ DQN considering that it is lightweight and sample-efficient. DoubelDQN and DuelDQN are two typical improved versions of DQN (Van Hasselt et al., 2016; Wang et al., 2016).

**Federated Learning.** In general, the optimization problem of federated learning (FL) can be formulated as:

$$\min_{w \in \mathbb{R}^d} f(w) := \frac{1}{m} \sum_{i=1}^{m} F_i(w), \quad (4)$$

where $F_i(w) \triangleq E_{\xi \sim D_i}[F_i(w, \xi)]$ represents the local loss function of the $i$-th client with the data sample $\xi$ drawn from distribution $D_i$. Data is typically heterogeneous for FL such that $D_i$ and $D_j$ can be extremely different for two distinct clients $i$ and $j$. Additionally, the FL systems often operate under a limited bandwidth and constrained client-side computing resources, making both the communication overhead from exchanging model parameters and the computational burden on the client a significant bottleneck.

Table 1: Classification accuracy (%) by altering bitwidths from INT4 to INT8 or from INT8 to INT4 on CIFAR-10.

|  | LeNet | ResNet-18 | MobileNetV2 |
|---|---|---|---|
| INT4→INT8 | 82.67 | 83.76 | 85.19 |
| INT8→INT4 | 82.30 | 82.81 | NaN |

## 4. Proposed Method

In this section, we first introduce our motivations, and then propose the low-bit integerized training framework for federal learning (LBI-FL), and finally elaborate on the training process of the temporal bit-width selection agent.

### 4.1. Motivation

Communication and computational costs are the two main costs in FL. Low-bit training can reduce the computational load during training, while the calculation with low bit-width weights helps mitigate both of these costs. However, low-bit integerized federated learning needs to address two challenges as summarized below.

**i) Varying impacts of quantization at different stages of training.** We conduct experiments on the CIFAR-10 dataset by switching from INT4 to INT8 at 40% of the training process and from INT8 to INT4 at 60% of the training process, respectively. Table 1 shows that using lower bit-widths in the later stage of training suffers from an evident performance loss, and might cause a model collapse. Therefore, it is meaningful to use different bit-widths at different training stages.

**ii) Varying training curves for different datasets and models.** Figure 2 shows that the quantization loss calculated by Equation (5) in (Zhang et al., 2020) varies across different models. We are thus motivated to train an RL agent to dynamically determine the temporal bit-width allocation for different models at different stages of training.

### 4.2. Low-Bit Integerized Federal Learning

As depicted in Figure 3, the proposed framework for low-bit integerized federal learning consists of two parts. First, the temporal bit-width selection agent is trained with reinforcement learning. The pretraining is conducted on a relatively small local dataset (10% of CIFAR-10 used in our approach). Second, the pre-trained agent is distributed to the clients to perform federated learning on new models or datasets. For every $Itv$ epochs, the agent on each client decides whether to adjust the bit-widths of weights, activations, and gradients.

The low-bit integerized federal training framework is detailed in Algorithm 1. This framework offers two main benefits for the acceleration of federated learning. (i) Since the

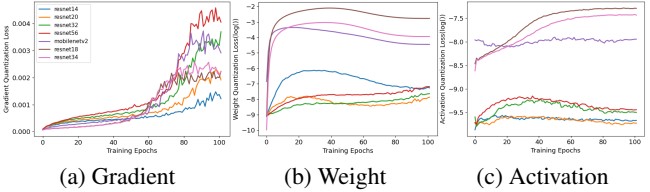

| (a) Gradient | (b) Weight | (c) Activation |
|---|---|---|

Figure 2: Average quantization loss of full INT8 quantization on (a) gradients, (b) weights, and (c) activations, respectively, over all the layers through the training process.

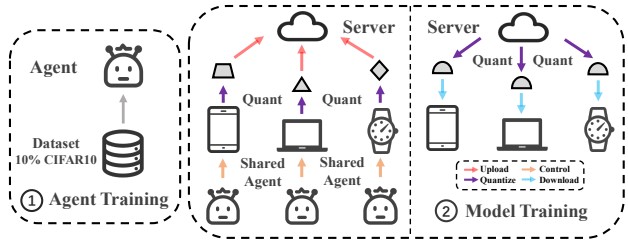

Figure 3: Overview of the proposed low-bit integerized federated learning. The agent is firstly pre-trained on a small local dataset, such as 10% of the CIFAR-10 dataset. It is then applied to another model or dataset for federated learning with low-bit training, which can reduce both the communication overhead and computational cost. The purple arrows, representing the uplinks and downlinks for communication, transmit low-bit weights with bit-width $w_{bit}^k$, leading to a significantly reduction in the communication cost.

model weights are quantized with low bit-widths, only the low-bit weights are transmitted when the model is uploaded from the clients to the server. Similarly, when the model is downloaded from the server, the weight is quantized before transmission. As a result, the communication overhead during both the uploading and downloading process is significantly reduced compared to the full-precision communication. (ii) Moreover, we implement a low-bit training on each client, replacing the full-precision matrix operations with low-bit integer ones, which can further reduce the computation cost and accelerate the training process.

### 4.3. Agent Training for Temporal Bit-Width Allocation

As illustrated in Figure 4, in our framework, temporally dynamic bit-width allocation is achieved with reinforcement learning. The agent is trained to determine whether to alter the bit-widths for the weights, activations, and gradients, considering the training state of the networks including training time and quantization loss of data. After each action, a reward is obtained based on the current model training accuracy and the compression ratio. The reward function,

**Algorithm 1** Low-bit Integerized Federated Learning.

**Input**: Pretrained agent, number of clients $M$, number of epochs $N$, and update interval Itv.

**Output**: Trained model with low-bit weights and activations.

1: Initialize the weight, activation, and gradient bit-widths $w_{bit}$, $a_{bit}$, and $g_{bit}$ for each client as INT4.
2: **for** $i = 1$ to $N$ **do**
3:      Train and update the local model with low bit-width $w_{bit}^k$, $a_{bit}^k$, and $g_{bit}^k$ on each client.
4:      Quantize the model on each client with bit-width $w_{bit}^k$ ($k \in [1, M]$) before uploading to the server.
5:      Server averages models to obtain a new global model.
6:      Quantize the model with bit-width $w_{bit}^k$ before downloading the model from the server to each client.
7:      **if** $i\%\text{Itv} == 0$ **then**
8:          Utilize the pre-trained agent to determine whether each client should adjust its bit-widths.
9:      **end if**
10: **end for**

therefore, needs to balance the maintenance of model accuracy and the reduction of computation cost. Since the early stage of training can tolerate more noise as shown in Section 4.1, the bit-widths are initialized as INT4 and can only switch from lower bit-widths to higher low-widths. Details of the reinforcement learning method are elaborated below.

**State space.** To well transfer the agent to different models and datasets, we exclude factors that are strongly related to the models and datasets. Consequently, the designed state space is composed of four types of factors.

(i) Time step that denotes the ratio of the current training epoch to the total number of epochs.
(ii) Current bit-width of weights, activations, and gradients.
(iii) Quantization loss of weights, activations, and gradients. The quantization loss is calculated using Equation (5) in (Zhang et al., 2020) to capture the data distribution difference for revealing the training process.
(iv) Last changing time step of the bit-width for weights, activations, and gradients. If the bit-width has never changed, the value is set to $-1$. From this value, the agent can estimate the compression ratio,

$$\text{Loss} = \left| \frac{\sum_i^n |x_f| - \sum_i^n |x_q|}{\sum_i^n |x_f|} \right|. \quad (5)$$

**Action space.** We adopt the mixed INT4, INT6 and INT8 format during the training stage. We set the agent only able to increase bit-widths. The agent has four actions: (i) Do not change the bit-widths. (ii-iv) Increase the bit-width of weights, activations, or gradients by $\Delta = 2$.

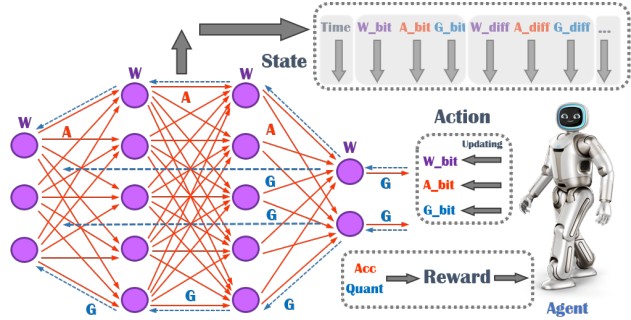

Figure 4: Overview of the reinforcement learning method of selecting bit-widths for the network during training.

**Balanced reward function.** As the agent aims to preserve accuracy and reduce the bit-width simultaneously, the reward function needs to balance the model accuracy and compression ratio. The balanced reward formulation is carefully designed, which consists of three parts: the quantization index, accuracy index, and time step.

$$\text{reward} = q\_idx \cdot a\_idx \cdot t\_step \quad (6)$$

*Quantization index.* In this paper, the computation cost of convolution is approximately calculated with BitOPs (Yang & Jin, 2021). Supposing the multiplication of a $k_w$ bit weight value and a $k_a$ bit activation value, the BitOPs of this multiplication is $k_w k_a$. BitOPs is an easy-to-calculate metric and is independent of the actual hardware.

The compression ratio $\lambda$ is obtained by comparing the current training with full INT8 quantization. The quantization index is formulated in Equation (7), where $\theta$ is the hyperparameter to adjust the quantization importance in the reward,

$$q\_idx = \theta/(1 - \lambda) \quad (7)$$

*Accuracy index.* To evaluate the model accuracy of low bit training, the model accuracy of full INT8 training at each epoch is stored as reference $a_r$. The accuracy index is obtained by comparing the current training accuracy with full INT8 model. $\delta$ in Equation (8) is also a hyperparameter to adjust the accuracy importance,

$$a\_idx = \begin{cases} 1 & \text{if} \quad a > a_r \\ 1/(1 + (a_r - a) * \delta) & \text{if} \quad a \leq a_r \end{cases} . \quad (8)$$

*Time step.* The ultimate objective of the agent is to increase the final compression ratio while maintaining the model accuracy. Therefore, the reward at the late stage of training is more important for which the reward is multiplied by the time step.

**Value network.** To keep the extra computation cost of the agent low during training, the agent network should be

kept small. Our Q-network has only one hidden layer of 128 neurons and the number of neurons in the first and final layer is equal to the number of states and the actions, respectively.

Since we do not have access to all the data in many situations, we train the agent on a relatively small local dataset (10% of the CIFAR-10) with ResNet-20 model and transfer it to other models and datasets.

### 4.4. Teacher Samples for Behaviour Cloning

As pure policy-based method (Schulman et al., 2017) usually needs the whole episode and the episode number in this scenario is limited, pure policy-based method is not suitable. Therefore, we adopt the typical off-policy method: deep Q-network (Mnih et al., 2015) to utilize its sample efficiency.

Behaviour cloning as a regularization (Goecks et al., 2020; Nair et al., 2018) is an effective tool to accelerate reinforcement learning from samples. Based on the observation in (Ding et al., 2024) that gradient and activation quantization have more impact on the model performance, we accumulate some state-action-reward pairs from this prior knowledge as the teacher samples $D_T$. Specifically, the possible teacher samples have gradient bit-width switched to INT8 with a time step ranging from $0.10$ to $0.45$ and the activation bit-width switched to INT8 with a time step from $0.50$ to $0.90$ and the changing time step of weight bit-width ranges from $0.50$ to $0.95$ or keeps at INT4. These samples can achieve a better balance between model performance and compression ratio than the random bit-width allocation.

The Q-network is trained on the teacher samples by supervised learning. To push the agent favored with the teacher sample's action, only calculating the value differences is not enough. We use the softmax function on the action dimension and utilize KL divergence to measure the distance between the agent policy and the accumulated teacher samples. Thus, the training loss is calculated as sum of value differences and the KL divergence between the agent policy and the teacher samples. Equation (9) is the training loss that makes the initial agent close to the teacher samples, where $\pi_T(s)$ is the teacher policy from the accumulated sample,

$$L_\theta^{BC} = \mathbf{E}_{(s,a) \in D_T} \left[ \left( \sum_{t=1}^M r(s_t = s, a_t = a) - q(s, a; \theta) \right)^2 \right]$$
$$+ \mathbf{E}_{s \in D_T} [\mathrm{KL}(\pi_T, \pi_\theta)], \quad (9)$$

where $\pi_\theta(a|s) = \exp(q(s, a; \theta)) / \sum_{a \in A} \exp(q(s, a; \theta))$ denotes the softmax of Q-values for different actions, $M$ is the episode length. By utilizing the teacher samples in the behavior cloning, the agent can be trained quickly in our scenario.

### 4.5. Convergence Analysis

In this section, we focus on the convergence analysis of the gradient quantization, considering that the impact of weight and activation quantization on the loss function could be accumulated by layers and is difficult to analyze quantitatively. Similar to Yang et al. (2021), we introduce Assumption 4.1 on the loss functions $f$ and $F_i$, $i = 1, \cdots, m$.

**Assumption 4.1.** Given any client $i \in M$ in the client set $M$, for $\mathbf{w}_t, \mathbf{w}'_t \in \mathbb{R}^d$, we have the following assumptions.

i) If $F$-function is *L-smooth*, then $\|\nabla F_i(\mathbf{w_t}) - \nabla F_i(\mathbf{w}'_t)\| \leq L\|\mathbf{w} - \mathbf{w}'_t\|$.

ii) Let $\xi_i$ be a random local data sample in the $t$-th step at the $i$-th worker, the local gradient estimator is unbiased $\mathbb{E}[\nabla F_i(\mathbf{w}_t, \xi_i)] = \nabla F_i(\mathbf{w}_t)$.

iii) The variance $\sigma_L$ of each local gradient estimator is bounded by $\mathbb{E}[\|\nabla F_i(\mathbf{w}_t, \delta_t^i) - \nabla F_i(\mathbf{w}_t)\|^2] \leq \sigma_L^2$, and the global variance $\sigma_G$ is bounded by $\|\nabla F_i(\mathbf{w}_t) - \nabla f(\mathbf{w}_t)\|^2 \leq \sigma_G^2$.

We further assume in Assumption 4.2 the equivalent additive noise for gradient quantization at each client.

**Assumption 4.2.** The gradient quantization noise on each client has a well-defined expectation and variance (denoted as $\mu_N^i$ and $(\sigma_N^i)^2$ for the $i$-th client), and the quantization noise expectation $\mu_N^i$ can be viewed as zero.

In Theorem 4.3, we develop the convergence rate for LBI-FL (with gradient quantization) associated with the total number of communication rounds $T$, and demonstrate that the theoretical convergence rate is equivalent to the rate of existing FL algorithms like FedAvg.

**Theorem 4.3.** *Under Assumptions 4.1 and 4.2 and full worker participation, when the learning rate $\eta \leq (8LK)^{-1}$ for each client, the output $w_t$ generated with gradient quantization satisfies:*

$$\min_{t \in [T]} \mathbb{E}\left[\|\nabla f(\mathbf{w}_t)\|_2^2\right] \leq \frac{f_0 - f_*}{c\eta KT} + \Phi, \quad (10)$$

*where $T$ is the number of communication rounds and $\Phi = \frac{1}{c}\left[\frac{L\eta}{2m}\left(\sigma_L^2 + \frac{1}{m}\sum_{i=1}^m\left((\mu_N^i)^2 + (\sigma_N^i)^2\right)\right) + \frac{5K\eta^2 L^2}{2}\left(\sigma_L^2 + 6K\sigma_G^2\right)\right]$, with $c$ as a constant, $K$ as the number of local updates, and $m$ as the number of clients.*

*Proof.* Please refer to Appendix A. □

Theorem 4.3 implies that, given $\eta = (\sqrt{T}LK)^{-1}$, the convergence rate of gradient quantization is $\mathcal{O}(1/\sqrt{T})$ when $T$ is sufficiently large, and is equivalent to the convergence rate of the standard FedAvg algorithm. In Section 5.4, we verify the convergence rate of LBI-FL with empirical evaluations.

# 5. Experiments

In this section, we first introduce the agent training result and then carry out extensive experiments on different models and datasets. Finally, ablation studies are conducted to evaluate the proposed method.

## 5.1. Agent Training

**Experiment setting.** We collect the teacher samples by training ResNet-20 (He et al., 2016) for 100 epochs on a subset of the CIFAR-10 dataset randomly sampled with 10% of the original data. In the reward function, $\theta$ is set as 0.25 and $\delta$ as 0.5. 200 episodes of samples are collected. The scaling parameters for quantization are updated every 100 iterations. The experiments in this paper are conducted on a single NVIDIA 3090 GPU.

**Training result.** The training agent converges where the gradient bit-width switches to high bit-widths in the very early stage and the activation bit-width switches based on the quantization difference during training while the weight bit-width fixes at INT4. Therefore, the communication overhead is reduced to **1/8** of the full-precision training method. The learned agent can achieve a good balance between compression ratio and model performance. In a new training scenario, we can directly use this agent or train a new agent with a small subset of the local data. Moreover, the agent consists of two linear layers with only 1.92K parameters, and requires 1.97G BitOPs for making one decision. The RL agent makes decision for every 5 epochs. Therefore, the computation cost is much smaller than the network training.

## 5.2. Evaluation Metrics

In our experiment, the communication cost reduced ratio and BitOPs reduced ratio (RR) compared to INT8 training, and the accuracy (Acc) are used as evaluation metrics. As the agent keeps weights at INT4 during the training process, the reduction ratio of communication cost is **87.5%** compared to full precision and **50%** compared to INT8 training. Below, we only report the results for Acc and BitOPs RR. For simplicity, the BitOPs of weight-activation convolution in the forward propagation are approximately treated the same as the gradient-weight convolution and gradient-activation convolution. For example, as the forward propagation of LeNet on CIFAR-10 is 14.81M Mac, the BitOPs under full precision are 14.81M$\times$32$\times$32$\times$3 = 45.5G.

## 5.3. Image Classification

**Experiment setting.** We evaluate with LeNet (LeCun et al., 1998), ResNet-18/50/101 (He et al., 2016), MobileNetV2 (Sandler et al., 2018), and ViT-S (Dosovitskiy et al., 2021) on the CIFAR-10/100 dataset. The agent deter-

Table 2: Top-1 classification accuracy (%) on CIFAR-10.

| Model | Method | Top-1 Acc (%) | BitOPs | RR (%) |
|---|---|---|---|---|
| LeNet | FP32 | 83.36 | 45.5G | - |
| | UI4 | 81.97 | 0.71G | 75 |
| | UI8 | 82.91 | 2.84G | 0 |
| | LBI-FL | 81.79 | 1.38G | 51.49 |
| ResNet-18 | FP32 | 84.21 | 114.2G | - |
| | UI4 | 82.71 | 1.78G | 75 |
| | UI8 | 84.23 | 7.14G | 0 |
| | LBI-FL | 84.16 | 3.28G | 54.06 |
| ResNet-50 | FP32 | 85.19 | 258.4G | - |
| | UI4 | 77.90 | 4.04G | 75 |
| | UI8 | 84.93 | 16.15G | 0 |
| | LBI-FL | 83.52 | 8.45G | 47.68 |
| ViT-S | FP32 | 73.85 | 1910.5G | - |
| | UI4 | 69.33 | 29.9G | 75 |
| | UI8 | 71.85 | 119.4G | 0 |
| | LBI-FL | 72.55 | 60.3G | 49.51 |
| MobileNet-V2 | FP32 | 92.04 | 290.6G | - |
| | UI4 | 85.83 | 4.54G | 75 |
| | UI8 | 89.65 | 18.2G | 0 |
| | LBI-FL | 89.02 | 8.83G | 51.39 |

Table 3: Top-1 classification accuracy (%) on CIFAR-100.

| Model | Method | Top-1 Acc (%) | BitOPs | RR (%) |
|---|---|---|---|---|
| Lenet | FP32 | 48.55 | 45.5G | - |
| | UI4 | 44.83 | 0.71G | 75 |
| | UI8 | 46.91 | 2.85G | 0 |
| | LBI-FL | 47.41 | 1.39G | 50.98 |
| ResNet-18 | FP32 | 54.35 | 114.4G | - |
| | UI4 | 49.35 | 1.79G | 75 |
| | UI8 | 53.57 | 7.15G | 0 |
| | LBI-FL | 53.48 | 3.30G | 53.81 |
| ViT-S | FP32 | 51.71 | 1910.5G | - |
| | UI4 | 46.39 | 29.9G | 75 |
| | UI8 | 49.32 | 119.4G | 0 |
| | LBI-FL | 48.32 | 57.2G | 52.08 |
| MobileNet-V2 | FP32 | 74.52 | 291.0G | - |
| | UI4 | 60.31 | 4.55G | 75 |
| | UI8 | 72.37 | 18.2G | 0 |
| | LBI-FL | 71.61 | 8.84G | 51.40 |

mines the bit-widths of weights, activations, and gradients from **INT4**, **INT6**, and **INT8** for every 5 epochs from the 10th epoch. For LeNet, the number of training epochs is set at 2000 and the client number is 100. 10% of the clients are selected to update at every epoch. For other larger networks, the number of training epochs is set at 200 and the client number is 10. All the clients are updated at every epoch. The local update epoch is 2 and the learning rate decay is 1.

**Results.** Tables 2 and 3 show that, compared to UI8 (Zhu et al., 2020), the proposed LBI-FL achieves less than 1.5% accuracy loss with 50% reduced BitOPs in most cases (except for LeNet on CIFAR-10) for various network architectures like LeNet, ResNet-18, ViT-S, and MobileNet-V2 on

Table 4: Convergence rate of LBI-FL compared with full-precision, full INT4, and INT8 training. It lists the number of epochs needed to obtain a target accuracy.

| Model | Acc (%) | FP32 | UI8 | UI4 | LBI-FL |
|---|---|---|---|---|---|
| LeNet | 81 | 472 | 612 | 624 | 569 |
| ResNet-18 | 81 | 68 | 70 | 106 | 76 |
| ViT-S | 68 | 38 | 52 | 66 | 56 |
| MobileNet-V2 | 84 | 36 | 39 | 61 | 50 |

Table 5: Comparison of our method with random selection.

| Model | Method | Acc (%) | BOPS | RR (%) |
|---|---|---|---|---|
| LeNet | Random | 81.76 (-0.03) | 1.41G | 50.25 |
| | LBI-FL | 81.79 | 1.38G | 51.49 |
| ResNet-18 | Random | 83.02 (-1.14) | 3.39G | 52.5 |
| | LBI-FL | 84.16 | 3.28G | 54.06 |
| MobileNet-V2 | Random | 86.91 (-2.11) | 8.65G | 52.5 |
| | LBI-FL | 89.02 | 8.83G | 51.39 |

Table 6: The effectiveness of our proposed method under non-iid conditions (Dirichlet = 0.25 / 0.5).

| Dataset | Dirichlet | Method | Top-1 Acc (%) | RR (%) |
|---|---|---|---|---|
| CIFAR-10 | 0.25 | FP32 | 80.39 | - |
| | | UI4 | 75.79 | 75 |
| | | UI8 | 78.71 | 0 |
| | | LBI-FL | 78.56 | 52.03 |
| CIFAR-10 | 0.5 | FP32 | 81.27 | - |
| | | UI4 | 77.42 | 75 |
| | | UI8 | 78.97 | 0 |
| | | LBI-FL | 78.90 | 51.68 |
| CIFAR-100 | 0.25 | FP32 | 46.22 | - |
| | | UI4 | 43.78 | 75 |
| | | UI8 | 45.81 | 0 |
| | | LBI-FL | 46.09 | 51.07 |
| CIFAR-100 | 0.5 | FP32 | 46.73 | - |
| | | UI4 | 44.43 | 75 |
| | | UI8 | 46.33 | 0 |
| | | LBI-FL | 46.25 | 51.06 |

Table 7: Effect of local update epochs.

| Dataset | Lep | Dirichlet | Top-1 Acc (%) | RR (%) |
|---|---|---|---|---|
| CIFAR-10 | 2 | iid | 81.79 | 51.49 |
| | 2 | 0.25 | 78.56 | 52.03 |
| | 5 | iid | 81.62 | 52.03 |
| | 5 | 0.25 | 77.93 | 53.20 |
| CIFAR-100 | 2 | iid | 47.41 | 50.98 |
| | 2 | 0.25 | 46.09 | 51.07 |
| | 5 | iid | 45.49 | 51.46 |
| | 5 | 0.25 | 43.38 | 51.70 |

Table 8: Effects of the learning rate decay.

| Dataset | LD | Dirichlet | Top-1 Acc (%) | RR (%) |
|---|---|---|---|---|
| CIFAR-10 | 1.0 | iid | 81.79 | 51.49 |
| | 1.0 | 0.25 | 78.56 | 52.03 |
| | 0.998 | iid | 81.53 | 51.59 |
| | 0.998 | 0.25 | 77.49 | 52.15 |
| CIFAR-10 | 1.0 | iid | 47.41 | 50.98 |
| | 1.0 | 0.25 | 46.09 | 51.07 |
| | 0.998 | iid | 42.69 | 51.00 |
| | 0.998 | 0.25 | 42.06 | 51.01 |

Table 9: Effect of initialization seeds.

| Dataset | Model | Seed | Top-1 Acc (%) | RR (%) |
|---|---|---|---|---|
| CIFAR-10 | LeNet | 23 | 81.79 | 51.49 |
| | | 60 | 82.01 | 51.43 |
| | | 90 | 81.71 | 51.45 |
| | | 200 | 81.92 | 51.60 |
| CIFAR-10 | ResNet-18 | 23 | 84.16 | 54.06 |
| | | 60 | 84.59 | 54.05 |
| | | 90 | 84.33 | 53.80 |
| | | 200 | 84.49 | 53.98 |
| CIFAR-10 | ViT-S | 23 | 72.55 | 49.41 |
| | | 60 | 72.52 | 49.66 |
| | | 90 | 72.21 | 49.75 |
| | | 200 | 72.17 | 49.74 |
| CIFAR-10 | MobileNet-V2 | 23 | 89.02 | 51.39 |
| | | 60 | 89.11 | 51.40 |
| | | 90 | 89.38 | 51.43 |
| | | 200 | 89.44 | 51.41 |

CIFAR-10/100.

Morever, the storage overhead of trained network weights and BitOPs for inference are reduced by 50%.

## 5.4. Convergency analysis

In addition to the model performance, the convergence rate (the number of epochs required to reach a target accuracy) is also a critical metric in federal learning. Table 4 compares the convergence rate of LBI-FL with full-precision, INT4, and INT8 training to obtain a target accuracy. The dataset is selected as CIFAR-10.

From Table 4, it is evident that the convergence rate of our method lies between full INT4 and INT8 training in most cases, and is even slightly faster than INT8 on the LeNet. Considering that the communication overload is half of that in INT8 training, and the computational load is also reduced by more than half, the slight increase in the number of convergence epochs is acceptable.

## 5.5. Ablation Studies

In this section, we first validate the effectiveness of the agent for the temporal bit-width selection, and then conduct ablation studies on some parameters in our method.

**Effect of temporal bit-width selection.** We compare the

proposed method with a direct approach where the clients at INT4 are randomly selected with a similar reduced ratio. The dataset is selected as CIFAR-10. For LeNet, 33 clients (100 clients overall) are set at INT4, and for ResNet-18 and MobileNet-V2, 3 clients (10 clients overall) are set at INT4.

Table 5 demonstrates that on small networks like LeNet, the performance of random selection may be acceptable, but the loss of performance on larger networks like ResNet-18 and MobileNet-V2 is not ignorable.

**Effect of data distribution.** We consider both iid (independent and identically distributed) and non-iid data distributions. The parameters of non-iid Dirichlet distribution are 0.25 and 0.5 in our experiments. Table 6 shows that, compared to INT8 training, the proposed method yields less than 0.5% performance loss on non-iid ddata distributions.

**Effect of local update epochs.** Table 7 demonstrates the impact of the number of local update epochs. It shows that using a smaller number of local update epochs at 2 has a small impact on CIFAR-10, but can significantly improve model performance on CIFAR-100.

**Effect of learning rate decay.** Table 8 compares two different learning rate decay (LD). It demonstrates that a learning rate decay of 1 is better, with the superiority being more pronounced on the CIFAR-100 dataset.

**Effect of initialization seeds.** We verify the impact of initialization seeds on our method. Table 9 reports the accuracy (Acc) and reduced ratio (RR) for training LeNet, ResNet-18, ViT-S, and MobileNet-V2 on CIFAR-10 using different seeds. The proposed method is not obviously affected by initialization seeds and performs well with all the seeds.

## 6. Conclusions

This paper proposes a low-bit training method in the federated learning scenario. We propose a scheme for temporally dynamic bit-width adjustment for deep neural networks. Deep Q-Network is employed to search for bit-width allocation schemes. Our approach significantly reduces communication overhead and computational cost in federated learning compared to full-precision training and achieves a good balance between compression rate and model performance. Comprehensive experiments on different models and datasets verify the versatility of our method. This paper provides a new effective approach to reducing overhead in federated learning and is worthy of further research.

## Acknowledgements

This work was supported in part by the National Natural Science Foundation of China under Grant 62320106003, Grant U24A20251, Grant 62401357, Grant 62401366, Grant 62431017, Grant 62125109, Grant 62371288, Grant 62301299, Grant 62120106007, in part by the Program of Shanghai Science and Technology Innovation Project under Grant 24BC3200800, and in part by the Al Laboratory of United Automotive Electronic Systems (UAES) Co. under Grant 2025-3270.

## Impact Statement

This paper presents a work centered on advancing the machine learning field, specifically within the realm of federated learning. There are very few potential societal consequences of our work, and none of them must be specifically highlighted here.

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

# A. Proof of Theorem 4.3

Our derivation is basically similar to that in Appendix A in (Yang et al., 2021). For a better understanding, please refer to Appendix A in (Yang et al., 2021).

The average update over the clients is $\bar{\Delta}_t = \frac{1}{m} \sum_{i=1}^{m} (\Delta_t^i + n_t^i)$. Take expectation of $f(x_{t+1})$ at communication round $t$, we can have:

$$
\begin{aligned}
\mathbb{E}_t [f(\mathbf{w}_{t+1})] &\leq f(\mathbf{w}_t) + \langle \nabla f(\mathbf{w}_t), \mathbb{E}_t [\mathbf{w}_{t+1} - \mathbf{w}_t] \rangle + \frac{L}{2} \mathbb{E}_t \left[ \| \mathbf{w}_{t+1} - \mathbf{w}_t \|^2 \right] \\
&= f(\mathbf{w}_t) + \langle \nabla f(\mathbf{w}_t), \mathbb{E}_t [\bar{\Delta}_t + \eta K \nabla f(\mathbf{w}_t) - \eta K \nabla f(\mathbf{w}_t)] \rangle + \frac{L}{2} \mathbb{E}_t \left[ \| \bar{\Delta}_t \|^2 \right] \\
&= f(\mathbf{w}_t) - \eta K \| \nabla f(\mathbf{w}_t) \|^2 + \underbrace{\langle \nabla f(\mathbf{w}_t), \mathbb{E}_t [\bar{\Delta}_t + \eta K \nabla f(\mathbf{w}_t)] \rangle}_{A_1} + \underbrace{\frac{L}{2} \mathbb{E}_t \left[ \| \bar{\Delta}_t \|^2 \right]}_{A_2}. \quad (11)
\end{aligned}
$$

As the noise $n_t^i$ from the gradient quantization has a well-defined expectation which can be viewed as zero (*Assumption 4*), we have $\mathbb{E}_t[n_t^i] = 0$. The term $A_1$ is calculated similar to (Yang et al., 2021):

$$
\begin{aligned}
A_1 &= \langle \nabla f(\mathbf{w}_t), \mathbb{E}_t [\bar{\Delta}_t + \eta K \nabla f(\mathbf{w}_t)] \rangle \\
&= \left\langle \nabla f(\mathbf{w}_t), \mathbb{E}_t \left[ -\frac{1}{m} \sum_{i=1}^{m} \sum_{k=0}^{K-1} \eta(\mathbf{g}_{t,k}^i + n_t^i) + \eta K \nabla f(w_t) \right] \right\rangle \\
&= \left\langle \nabla f(\mathbf{w}_t), \mathbb{E}_t \left[ -\frac{1}{m} \sum_{i=1}^{m} \sum_{k=0}^{K-1} \eta \mathbf{g}_{t,k}^i + \eta K \nabla f(w_t) \right] \right\rangle \\
&\leq \eta K \left( \frac{1}{2} + 15 K^2 \eta^2 L^2 \right) \| \nabla f(w_t) \|^2 + \frac{5 K^2 \eta^3 L^2}{2} (\sigma_L^2 + 6 K \sigma_G^2) - \frac{\eta}{2 K m^2} \mathbb{E}_t \left\| \sum_{i=1}^{m} \sum_{k=0}^{K-1} \nabla F_i(\mathbf{w}_{t,k}^i) \right\|^2. \quad (12)
\end{aligned}
$$

Moreover, the term $A_2$ can be bounded as:

$$
\begin{aligned}
A_2 &= \mathbb{E}_t \left[ \| \bar{\Delta}_t \|^2 \right] = \mathbb{E}_t \left[ \left\| \frac{1}{m} \sum_{i=1}^{m} (\Delta_t^i + n_t^i) \right\|^2 \right] \\
&\leq \frac{1}{m^2} \mathbb{E}_t \left[ \left\| \sum_{i=1}^{m} (\Delta_t^i + n_t^i) \right\|^2 \right] = \frac{\eta^2}{m^2} \mathbb{E}_t \left[ \left\| \sum_{i=1}^{m} \sum_{k=0}^{K-1} \mathbf{g}_{t,k}^i \right\|^2 \right] + \frac{\eta^2}{m^2} K \cdot \mathbb{E}_t \left[ \left\| \sum_{i=1}^{m} n_t^i \right\|^2 \right]. \quad (13)
\end{aligned}
$$

Considering the quantization noise on each client can be supposed independent, the expectation in the second term is calculated as:

$$
\mathbb{E}_t \left[ \left\| \sum_{i=1}^{m} n_t^i \right\|^2 \right] = \sum_{i=1}^{m} \mathbb{E}_t \left[ \| n_t^i \|^2 \right] + 2 \sum_{i<j} \mathbb{E}_t \left[ n_t^i n_t^j \right] = \sum_{i=1}^{m} \left( (\mu_N^i)^2 + (\sigma_N^i)^2 \right) + 2 \sum_{i<j} \mu_N^i \mu_N^j. \quad (14)
$$

According to the assumption that the quantization noise expectation $\mu_N^i$ and $\mu_N^j$ can be viewed as zero, we can have:

$$
\mathbb{E}_t \left[ \left\| \sum_{i=1}^{m} n_t^i \right\|^2 \right] = \sum_{i=1}^{m} \left( (\mu_N^i)^2 + (\sigma_N^i)^2 \right). \quad (15)
$$

Table 10: Ablation studies on training LeNet on CIFAR-10.

| Dirichlet | Lep | LD | CIFAR-10 | | CIFAR-100 | |
|---|---|---|---|---|---|---|
| | | | Top-1 Acc (%) | RR (%) | Top-1 Acc (%) | RR (%) |
| iid | 2 | 1 | 81.79(-1.12) | 51.49 | 47.41(+0.50) | 50.98 |
| | | 0.998 | 81.53(-0.57) | 51.59 | 42.69(-1.15) | 51.00 |
| | 5 | 1 | 81.62(-1.10) | 52.03 | 45.49(+1.23) | 51.46 |
| | | 0.998 | 81.73(-0.28) | 52.83 | 39.74(+0.62) | 51.31 |
| 0.25 | 2 | 1 | 78.56(+0.25) | 52.03 | 46.09(+1.28) | 51.07 |
| | | 0.998 | 77.49(-0.83) | 52.15 | 42.06(-1.08) | 51.01 |
| | 5 | 1 | 77.93(-1.41) | 53.20 | 43.38(-0.62) | 51.70 |
| | | 0.998 | 78.62(+0.10) | 53.74 | 40.00(-0.08) | 51.59 |
| 0.5 | 2 | 1 | 78.90(+0.13) | 51.68 | 46.25(-0.08) | 51.06 |
| | | 0.998 | 79.81(+0.09) | 51.91 | 44.53(-0.58) | 51.00 |
| | 5 | 1 | 80.05(-0.73) | 52.56 | 45.14(+0.51) | 51.56 |
| | | 0.998 | 79.83(-0.38) | 53.35 | 40.09(-0.51) | 51.44 |

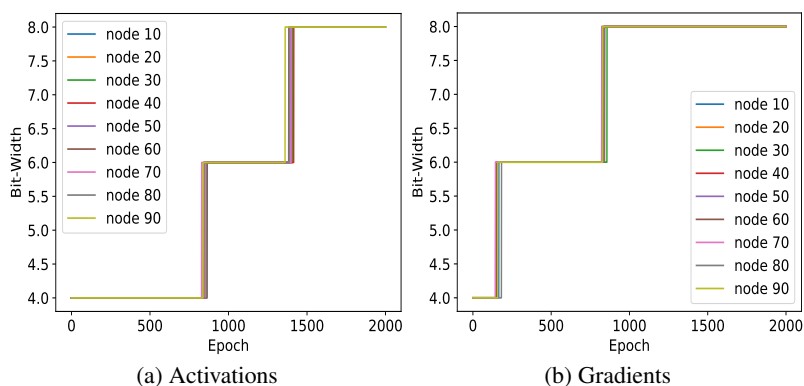

(a) Activations

(b) Gradients

Figure 5: The bitwidth change process (INT4 → INT6, INT6 → INT8) of (a) activations, and (b) gradients through training LeNet on CIFAR-10.

From $A_1$ and $A_2$, we can have:

$$\mathbb{E}_t \left[ f \left( \mathbf{w}_{t+1} \right) \right] \leq f \left( \mathbf{w}_t \right) - \eta K \left\| \nabla f \left( \mathbf{w}_t \right) \right\|^2 + \underbrace{< \nabla f \left( \mathbf{w}_t \right), \mathbb{E}_t \left[ \bar{\Delta}_t + \eta K \nabla f \left( \mathbf{w}_t \right) \right] >}_{A_1} + \frac{L}{2} \underbrace{\mathbb{E}_t \left[ \left\| \bar{\Delta}_t \right\|^2 \right]}_{A_2}$$

$$\leq f \left( \mathbf{w}_t \right) - c\eta K \left\| \nabla f \left( \mathbf{w}_t \right) \right\|^2 + \frac{LK\eta^2}{2m} \left( \sigma_L^2 + \frac{1}{m} \sum_{i=1}^m ((\mu_N^i)^2 + (\sigma_N^i)^2) \right) + \frac{5\eta K^2 \eta^3 L^2}{2} \left( \sigma_L^2 + 6K\sigma_G^2 \right). \tag{16}$$

Similarly to (Yang et al., 2021), rearranging and summing from $t = 0$ to $t = T - 1$, we have:

$$\min_{t \in [T]} \mathbb{E} \left[ \left\| \nabla f \left( \mathbf{w}_t \right) \right\|_2^2 \right] \leq \frac{f_0 - f_*}{c\eta KT} + \Phi, \tag{17}$$

where $\Phi = \frac{1}{c} \left[ \frac{L\eta}{2m} (\sigma_L^2 + \frac{1}{m} (\sum_{i=1}^m ((\mu_N^i)^2 + (\sigma_N^i)^2))) \right] + \frac{1}{c} \left[ \frac{5K\eta^2 L^2}{2} \left( \sigma_L^2 + 6K\sigma_G^2 \right) \right]$. As a result, Equation (10) is valid.

## B. Ablation Studies Results

Table 10 presents more comprehensive results of ablation studies under different data distributions and hyperparameters. The numbers in parentheses indicate the model performance loss compared to full INT8 training. It can be seen that our method performs well in iid and non-iid scenarios. In most cases, models with local update epochs (Lep) of 2 and learning rate decay (LD) of 1 perform better, which is consistent with what is mentioned in the main text.

## C. Bit-Width Change Process

Figure 5 illustrates the bit-width change process of activations and gradients during the training of LeNet on the CIFAR-10 dataset. The weights are maintained at INT4 precision during training. It can be observed that gradient bit-widths increase at an earlier stage, while activations increase in the later process. There are some differences among different clients, but these differences are not significant.

