# OpenReview forum: "LBI-FL: Low-Bit Integerized Federated Learning with Temporally Dynamic Bit-Width Allocation"
_ICML.cc/2025/Conference — ICML 2025 poster_

### Official Review · Reviewer_KpQQ · 2025-03-12

**Overall Recommendation:** 4

**Summary:**

The proposed method quantifies weights, activation, and gradients to lower accuracy than INT8 to significantly reduce communication and computational costs, and the proposed LBIFL can reduce communication costs by 8 times compared to full-precision FL. A large number of experiments show that compared with INT8 precision low-bit training, the proposed LBI-FL reduces the average BitOPs per client by more than 50% at a precision loss of less than 2%.

**Claims And Evidence:**

yes

**Essential References Not Discussed:**

For work on the combination of model compression and federated learning, using a combination of model pruning and quantification, the methods proposed in this paper focus on the application of quantitative methods in federated learning, but there may be differences in the comprehensiveness and efficiency of model compression compared with these methods combining pruning and quantification, which may not be fully discussed in this paper.

**Experimental Designs Or Analyses:**

The experimental design of the paper is basically reasonable, and the purpose of the experiment is to verify the effectiveness of the proposed method in reducing communication cost and computing cost. CIFAR-10 and CIFAR-100 data sets are selected for verification, and TOP-1 classification accuracy is adopted. The experiment mainly focuses on the image classification task, and the data set is relatively small, and the method comparison in this paper is not compared with some advanced compression methods.

**Methods And Evaluation Criteria:**

Experiments are conducted on CIFAR-10 and CIFAR-100 datasets, and it is suggested to add experiments on more different types of federated learning scenarios to verify the effectiveness and scalability of the method in different environments.

**Other Comments Or Suggestions:**

no

**Other Strengths And Weaknesses:**

In addition, Figure 1 of the article is not referenced elsewhere in the article.

**Questions For Authors:**

1.Can the source code be made public？
2.The low-bit training is well understand, is any difference with other methd, like in deep learning?

**Relation To Broader Scientific Literature:**

The main contributions of this paper are mainly in three areas: federated learning, model compression and reinforcement learning
Federated learning technology is used to protect privacy, but federated learning faces the limitation of communication and computing resources. Therefore, low-bit quantization technology is applied to federated learning to compress the model and reduce the model size and computing cost. Finally, reinforcement learning is applied to optimize the allocation of bit width. Reinforcement learning provides new scenarios in hyperparameter tuning and network architecture searching.

**Theoretical Claims:**

The theory in this paper is proved to be correct, the convergence of the framework is analyzed, and the convergence rate of the framework is deduced, which is proved to be equivalent to the standard FedAvg algorithm. At the same time, the quantization error is analyzed, considering the influence of the noise introduced by gradient quantization on the model updating and convergence, and the quantization error will not have a fundamental impact on the convergence.

---

> ### Author Rebuttal · Authors · 2025-04-01
>
> Thank you for your valuable suggestions and below are our detailed responses to the raised weaknesses and questions.
>
> > **W1: Experiments on different scenarios.**
>
> **Response:** We have made comprehensive evaluations on our LBI-FL, including image classification on CIFAR-10 using diverse architectures of ResNet-18/50/101, MobileNet-V2, and ViT-S and on Tiny-ImageNet using ResNet-18, and image segmentation on DSB2018 using U-Net. In all the evaluations, our LBI-FL achieves acceptable performance (tolerable 1.5\% loss) with over 45\% reduction of BitOPs, compared with UI8 training.
>
> | Model | Method | Acc (\%) | BitOPS | RR (\%) |
> | ---- | ---- | ---- | ---- | ---- |
> | ResNet50 | FP32 | 85.19 | 258.4G | - |
> | | UI4 | 77.90 | 4.04G | 75 |
> | | UI8 | 84.93 | 16.15G | 0 |
> | | LBI-FL | 83.52 | 8.45G | 47.68 |
> | ResNet101 | FP32 | 85.52 | 492.0G | - |
> | | UI4 | NaN | 7.69G | 75 |
> | | UI8 | 85.05 | 30.75G | 0 |
> | | LBI-FL | 83.55 | 16.31G | 46.96 |
>
>
> > **W2: Experiments on more tasks and datasets and comparison with advanced compression methods.**
>
> **Response:** i) We train U-Net for Image segmentation on DSB2018. We adopt the image size of 96$\times$96. The table below shows that, compared with UI8 training, we achieve acceptable performance of about 0.6\% loss in Dice similarity coefficient (DSC) with over 50\% reduction of BitOPs.
>
> | Dataset | Method | DICE (\%) | BitOPS | RR (\%) |
> | ---- | ---- | ---- | ---- | ---- |
> | DSB2018 | FP32 | 89.55 | 19848G | - |
> | | UI4 | 87.01 | 310.1G | 75 |
> | | UI8 | 89.48 | 1240.5G | 0 |
> | | LBI-FL | 88.84 | 604.9G | 51.27 |
>
> ii) We train ResNet-18 for image classification on Tiny-ImageNet. The table below shows that our LBI-FL achieves over 50\% reduction of BitOPs with less than 0.1\% accuracy loss
>
> | Dataset | Method | Acc (\%) | BitOPS | RR (\%) |
> | ---- | ---- | ---- | ---- | ---- |
> | Tiny-ImageNet | FP32 | 35.84 | 457.14G | - |
> | | UI4 | 34.64 | 7.14G | 75 |
> |  | UI8 | 35.21 | 28.57G | 0 |
> | | LBI-FL | 35.13 | 14.05G | 50.81 |
>
> iii) We further employ centralized low-bit training method AMPA [D1] in FL for comparison. AMPA uses layer-wise bit-width allocation based on sensitivity measurement during training. Our LBI-FL with RL agents obtains superior performance with evidently reduced BitOPs in training ResNet-18, MobileNet-V2, and ViT-S on CIFAR-10.
>
> [D1] Li Ding et al. AMPA: Adaptive mixed precision allocation for low-bit integer training. ICML 2024.
>
> | Model | Method | Acc (\%) | BitOPS | RR (\%) |
> | ---- | ---- | ---- | ---- | ---- |
> | ResNet-18 | AMPA | 84.10 | 3.49G | 51.11 |
> | | LBI-FL | 84.16 | 3.28G | 54.06 |
> | ViT-S | AMPA | 72.54 | 73.08G | 38.79 |
> | | LBI-FL | 72.55 | 60.3G | 49.51 |
> | MobileNet-V2 | AMPA | 87.89 | 9.78G | 46.19 |
> |  | LBI-FL | 89.02 | 8.83G | 51.39 |
>
> > **W3: Discussion on combining pruning and quantification.**
>
> **Response:** This paper focuses on quantization for low-bit training in FL. Our LBI-FL introduces a temporally dynamic bit-width allocation scheme for weights, activations, and gradients, which evolves along the training trajectory. It achieves higher efficiency than UI8 training with great flexibility in diverse FL scenarios.
>
>
> We will discuss the meaningful topic of combining pruning and quantization such as [D2] in the final version. However, pruning is more difficult to deploy on hardware than quantization and relies on manually tuned hyperparameters such as pruning ratios and threshold values. [D2] only offers results of LeNet and ResNet-20 on CIFAR-10. We will explore to combine pruning with low-bit training (simultaneously quantizing weights, activations and gradients) in future.
>
> [D2] Pavana Prakash et al. IoT device friendly and communication-efficient federated learning via joint model pruning and quantization. IEEE IoT Journal 2022.
>
> > **W4: Figure 1 not referenced.**
>
> **Response:** Thank you. We will include the reference to Figure 1 in the final version.
>
> > **Q1: Can the source code be made public?**
>
> **Response:** We will release the source code upon acceptance.
>
> > **Q2: Difference with other methods like in deep learning.**
>
> **Response:** While low-bit training has been widely studied in conventional deep learning, its direct application to federated learning (FL) is challenging due to the decentralized and heterogeneous nature of FL. In particular, applying a uniform low-bit strategy across all clients can lead to training instability caused by heterogeneity in data distributions and model dynamics. To address this, we propose a aware bit width adaptation framework based on reinforcement learning. Specifically, we introduce an agent that dynamically allocates bit-widths for each client by considering local states, including the current bit-width, training phase, and quantization-induced loss.

---

> > ### Comment · Reviewer_KpQQ · 2025-04-05
> >
> > Thank you for your improvement.

---

> > > ### Author Response · Authors · 2025-04-06
> > >
> > > Dear Reviewer KpQQ,
> > >
> > > We greatly appreciate your constructive suggestions, which have helped us a lot to improve our paper.
> > >
> > > Thank you again for dedicating your time and effort to reviewing our paper and providing insightful comments.
> > >
> > > Best regards,
> > > The Authors

---

### Official Review · Reviewer_umsU · 2025-03-14

**Overall Recommendation:** 3

**Summary:**

The paper introduces Low-Bit Integerized Federated Learning (LBI-FL), a framework designed to reduce both communication and computational costs in Federated Learning (FL) by using quantization techniques. Unlike conventional approaches limited to INT8 precision, LBI-FL dynamically adjusts the bit-width of weights, activations, and gradients during training through reinforcement learning. A trained agent optimizes bit-width allocation by considering factors such as current precision, training stage, and quantization loss. The method generalizes well across different network architectures and non-IID datasets. Authors claim that theoretical analysis confirms that gradient quantization maintains the same convergence rate as FedAvg. Experiments show that LBI-FL achieves 8× communication cost reduction and over 50% fewer BitOPs per client with less than 2% accuracy loss compared to INT8-based low-bit training.

**Claims And Evidence:**

In this paper, the authors make several claims that may raise concerns.

The first statement, found in lines 14-17 on the left side of page 1, reads:
*"Existing compression methods for FL cannot simultaneously reduce the up-link and downlink communication cost and mitigate the computation burden on clients."*
This claim is problematic, as there is at least one paper—Meinhardt, Georg, et al. (2024), *"Prune at the Clients, Not the Server: Accelerated Sparse Training in Federated Learning"*—that addresses the simultaneous reduction of both communication costs and computational load through client-side pruning.

Another concerning claim appears in the contribution section:
*"To our best knowledge, this is the first successful attempt to achieve low-bit training FL that evidently reduces the communication overhead and computation cost compared to full-precision and INT8 training."*
This assertion should be carefully justified, as there is at least one paper addressing low-bit training in distributed setups, which is not referenced here. Therefore, calling it the "first successful attempt" seems unsupported.

Mishchenko, Konstantin, et al. "IntSGD: Adaptive floatless compression of stochastic gradients." arXiv preprint arXiv:2102.08374 (2021).



A further troubling statement is:
*"We demonstrate in theory that federated learning with gradient quantization achieves an equivalent convergence rate to the standard FedAvg algorithm with sufficiently large number of communication rounds and further empirically verify the convergence rate."*
This analysis, found in the contribution section, heavily relies on Assumption 4.2, which states that gradient quantization at each client is equivalent to introducing Gaussian noise. This assumption is quite restrictive and not realistic in practical scenarios. In contrast, the analysis of FedAvg does not rely on assumptions about the distribution, making the comparison of convergence rates less valid and the statement potentially misleading.

**Essential References Not Discussed:**

As I mentioned earlier, there are several papers that have not been included:

Meinhardt, Georg, et al. (2024), *"Prune at the Clients, Not the Server: Accelerated Sparse Training in Federated Learning"

Mishchenko, Konstantin, et al. "IntSGD: Adaptive floatless compression of stochastic gradients." arXiv preprint arXiv:2102.08374 (2021).

**Experimental Designs Or Analyses:**

The authors evaluate the proposed method using LeNet (LeCun et al., 1998), ResNet-18 (He et al., 2016), MobileNetV2 (Sandler et al., 2018), and ViT-S (Dosovitskiy et al., 2020) on the CIFAR-10/100 dataset. The agent selects the bit-widths for weights, activations, and gradients from INT4, INT6, and INT8 every five epochs, starting from the 10th epoch. For LeNet, the training process consists of 2000 epochs with 100 clients, where 10% of the clients are selected for updates in each epoch. For the larger networks, training is conducted over 200 epochs with 10 clients, all of whom participate in updates at every epoch. The local update epoch is set to 2, and the learning rate follows a decay of 1.

I appreciate the detailed experimental evaluation; however, conducting experiments on larger models could further strengthen the study. Quantization has a particularly significant impact on large-scale models, where computational and communication efficiency are critical factors. Evaluating the proposed method on more complex architectures would provide deeper insights into its scalability and effectiveness in real-world scenarios.

===========================================

After rebuttal:
Authors provided additional results, which is valuable.

**Methods And Evaluation Criteria:**

The proposed method appears to be well-founded, with the authors employing both theoretical analysis and experimental results to evaluate its effectiveness.

**Other Comments Or Suggestions:**

Please review the previous sections.

**Other Strengths And Weaknesses:**

N/A

**Questions For Authors:**

Is it possible for you to obtain convergence results without Assumption 4.2, or with a relaxed version of it?

**Relation To Broader Scientific Literature:**

The contributions presented in this paper are relevant to the broader Federated Learning literature, particularly in the context of communication compression in Federated Learning.

**Theoretical Claims:**

I have carefully reviewed the proof presented in the appendix. The analysis relies extensively on Assumption 4.2, which essentially reduces the complexity of the proof, making the analysis overly simplistic and trivial. This reliance on such a restrictive assumption undermines the robustness of the theoretical framework. Specifically, the assumption that gradient quantization is equivalent to introducing Gaussian noise at each client is unrealistic and does not reflect the practical dynamics of Federated Learning. Consequently, the current analysis fails to provide a meaningful or comprehensive theoretical understanding of the method's behavior in real-world scenarios. The overly simplistic nature of the proof, due to this assumption, limits its applicability and generalizability to more complex or realistic settings.

===========================================

After rebuttal:

Authors provided the updated analysis as requested, so I increased the score accordingly.

---

> ### Author Rebuttal · Authors · 2025-04-01
>
> Thank you for your constructive suggestions.
>
> > **W1: Claims may raise concerns.**
>
> **Response:** Claim i): We use this sentence to emphasize that our LBI-FL can reduce both the overhead in the uplink and downlink communication with low-bit training. In fact, the paper "Prune at the Clients, Not the Server" cannot reduce the downlink communication only using client-side pruning and has to resort to accelerated server pruning, which could completely fail in accuracy and loss as mentioned in the paper. For rigidity, we will change the claim to "existing methods based on quantization" in the final version.
>
> Claim ii): In lines 19-21 and 55-59, "low-bit training" refers to simultaneously quantizing the weights, activations, and gradients. The paper "IntSGD: Adaptive floatless compression of stochastic gradients" develops adaptive integer compression operators for distributed SGD. It only quantizes the gradients and does not quantize weights and activations. To our best knowledge, we are the first to achieve "low-bit training" in the FL scenario.
>
>
> Claim iii): We have improved the theoretical result by relaxing Assumption 4.2 without affecting the final convergence rate. Instead of assuming the quantization noise to follow a Gaussian distribution with identical variance across all the clients, we now assume only that (a) The gradient quantization noise on each client has a well-defined expectation and variance ($\mu_N^i$ and $(\sigma_N^i)^2$ for the $i$-th client), and (b) The expectation $\mu_N^i$ can be viewed as zero.
>
> Based on this assumption, we provide an updated derivation of the theorem. The noise term does not influence the $A_1$ term in Equation (12). Neglecting the differences across epochs, we reformulate Equation (13) in the appendix as:
>
> $$
> \begin{aligned}
> A_2
> & \le \frac{\eta^2}{m^2} \mathbb{E}\_t\left[\left\|\sum\_{i=1}^m \sum_{k=0}^{K-1} \mathbf{g}\_{t, k}^i\right\|^2\right] +
> \frac{\eta^2}{m^2} K \cdot \mathbb{E}\_t \left[ \left\| \sum\_{i=1}^m n_t^i \right\|^2\right]
> \end{aligned}
> $$
>
> Considering the quantization noise on each client can be supposed independent, the expectation in the second term is calculated as:
>
> $$
> \mathbb{E}\_t \left[ \left\| \sum\_{i=1}^m n_t^i \right\|^2\right]  = \sum\_{i=1}^m \mathbb{E}\_t \left[ \left\|  n_t^i \right\|^2\right] + 2\sum_{i<j} \mathbb{E}_t\left[n_t^i n_t^j\right] = \sum\_{i=1}^m ((\mu\_N^i)^2 + (\sigma^i\_N)^2) + 2\sum\_{i<j} \mu_N^i \mu_N^j
> $$
>
> Therefore, $\Phi$ in Equation (10) changes as
>
> $$
> \Phi = \frac{1}{c}\left[\frac{L \eta}{2 m} \left(\sigma_L^2 + \frac{1}{m} \left( \sum\_{i=1}^m  ((\mu\_N^i)^2 + (\sigma^i\_N)^2)  \right)\right)   +\frac{5 K \eta^2 L^2}{2}\left(\sigma_L^2+6 K \sigma_G^2\right)  \right]
> $$
>
> However, the convergence rate does not change, since the quantization noise expectation is usually at 0.
>
> Note that previous convergence result is a special case of this new result.
>
> > **W2: Assumption for theoretical claims.**
>
> **Response:** We perform experiments with ResNet-18 on CIFAR-10 to support the assumption (b) in W1 that the expectation $\mu_N^i$ can be viewed as zero. The distribution of quantization noise is shown to concentrate around zero.
>
> > **W3: Evaluations on more complex architectures.**
>
> **Response:** We have evaluated on larger-scale ResNet50 and ResNet101 on CIFAR-10. The table below shows that our LBI-FL yields less than 1.5\% performance loss with over 45\% BitOPs reduction.
>
> | Model | Method | Dice (\%) | BOPS | RR (\%) |
> | ---- | ---- | ---- | ---- | ---- |
> | ResNet50 | FP32 | 85.19 | 258.4G | - |
> | | UI4 | 77.90 | 4.04G | 75 |
> | | UI8 | 84.93 | 16.15G | 0 |
> | | LBI-FL | 83.52 | 8.45G | 47.68 |
> | ResNet101 | FP32 | 85.52 | 492.0G | - |
> | | UI4 | NaN | 7.69G | 75 |
> | | UI8 | 85.05 | 30.75G | 0 |
> | | LBI-FL | 83.55 | 16.31G | 46.96 |
>
> We further evaluate our LBI-FL by training U-Net for image segmentation on DSB2018. The table below shows that, compared with UI8 training, our LBI-FL obtains about 0.6\% loss in Dice similarity coefficient (DSC) with over 50\% reduction of BitOPs.
>
> | Dataset | Method | Dice (\%) | BOPS | RR (\%) |
> | ---- | ---- | ---- | ---- | ---- |
> | DSB2018 | FP32 | 89.55 | 19848G | - |
> | | UI4 | 87.01 | 310.1G | 75 |
> | | UI8 | 89.48 | 1240.5G | 0 |
> | | LBI-FL | 88.84 | 604.9G | 51.27 |
>
> > **W4: References Not Discussed.**
>
> **Response:** We carefully read the two papers and will definitely cite them in the final version. However, they are different from our paper in the contents, as elaborated in W1.
>
> > **Q1: Convergence results without Assumption 4.2 or with a relaxed version.**
>
> **Response:** We have provided a relaxed version of Assumption 4.2. We only assume the expectation and variance of the distribution of the gradient quantization noise on each client exist and the expectation can be viewed as zero, as elaborated in W1. The relaxed assumption does not change the convergence rate in the theorem.

---

> > ### Comment · Reviewer_umsU · 2025-04-02
> >
> > Thank you to the authors for their detailed responses and the additional experiments provided. I appreciate the updated theoretical analysis, so I increase my score accordingly.
> >
> > Best regards,
> > Reviewer

---

> > > ### Author Response · Authors · 2025-04-03
> > >
> > > Dear Reviewer umsU,
> > >
> > > We sincerely appreciate the time and effort that you have dedicated to reviewing our paper and providing these insightful comments, which will further help improve the quality of the final version of our manuscript.
> > >
> > > Best regards,
> > > The Authors

---

### Official Review · Reviewer_Nft9 · 2025-03-24

**Overall Recommendation:** 4

**Summary:**

This paper proposes a low-bit integerized federated learning (LBI-FL) framework, which reduces the uplink and downlink communication overhead and mitigates the computation burden on clients, all with a tolerable level of performance loss. Specifically, a reinforcement learning based agent, which is trained on a small local dataset, is applied to dynamically determine the bit-widths for weights, activations, and gradients. The authors demonstrate, in theory, that federated learning with gradient quantization can achieve an equivalent convergence rate to the standard FedAvg algorithm with a sufficiently large number of communication rounds. Experimental results show the proposed framework 1) reduces the communication overhead to 1/8 of the full-precision training method; 2) reduces over 50% BitOPs per client on average with less than 2% accuracy loss, compared to INT8 training.

**Claims And Evidence:**

The authors verified the validity of the proposed scheme with comprehensive experiments.

**Essential References Not Discussed:**

One essential reference, which proposed the standard settings of federated learning and the framework of FedAvg, should be cited:

McMahan, Brendan, et al. "Communication-efficient learning of deep networks from decentralized data." Artificial intelligence and statistics. PMLR, 2017.

**Experimental Designs Or Analyses:**

The authors conducted experiments with the CIFAR-10/100 dataset and models of LeNet, ResNet-18, MobileNet-V2, and ViT-S and compared the communication and computation costs, BitOPs, and accuracy under various bit widths. They also checked the effects of local update epochs, data distribution, learning rate decay, etc.

**Methods And Evaluation Criteria:**

Yes. The authors leverage reinforcement learning to dynamically determine client bit-widths in federated learning, allowing more flexible and reasonable bit-widths allocation.

**Other Comments Or Suggestions:**

Typos:
1) In Section 4.2, "Every Thr epochs, the agent on each client ...".
2) In Section 4.3, the part of the Balanced reward function, "Supposng the multiplication of a ..."
3) In Section 4.5, "the theoretical convergence rate is equivalent to the rrate of ..."

**Other Strengths And Weaknesses:**

Strengths:
1) The core idea, leveraging reinforcement learning to determine the compression strategy in federated learning, is novel, reasonable, and potential.
2) Comprehensive experiments verify the validity of the proposed methodology.

Weaknesses:
1) The theoretical proof in the appendix is not rounded enough, though, as stated, it is similar to one reference.

## Update after rebuttal

The authors gave a rounded analysis as demanded, which makes the theory more generalized. Therefore, I increased the score.

**Questions For Authors:**

As for the bit-width change process, is it the same case (as LeNet on CIFAR-10) for other models on other datasets? (There are no significant differences among clients, but there is a difference between activations and gradients).

As for the proof, it would be better to provide clear claims for notations. Also, is it applicable to regard the quantization error as Gaussian noise under the high heterogeneity of federated learning?

**Relation To Broader Scientific Literature:**

This paper introduces reinforcement learning methodology into model compression under federated settings. This new approach reduces overhead in federated learning, and the idea could be explored more widely, for example, in personalized federated learning or channel-wise compression strategy.

**Theoretical Claims:**

Yes. However, the proof of the theorem in this paper is not rounded enough.

---

> ### Author Rebuttal · Authors · 2025-04-01
>
> Thank you for your valuable suggestions and below are our responses to the weaknesses and questions.
>
> > **W1: Proof of the theorem.**
>
> **Response:** According to your comment, we have improved the theoretical analysis by relaxing Assumption 4.2 without affecting the final convergence rate. Specifically, instead of assuming the quantization noise to follow a Gaussian distribution with identical variance across all the clients, we now assume only that (a) The gradient quantization noise on each client has a well-defined expectation and variance (denoted as $\mu_N^i$ and $(\sigma_N^i)^2$ for the $i$-th client), and (b) The quantization noise expectation $\mu_N^i$ can be viewed as zero.
>
> Based on this assumption, we provide an updated derivation of the theorem. The noise term does not influence the $A_1$ term in Equation (12). Neglecting the differences across epochs, we reformulate Equation (13) in the appendix as:
>
> $$
> \begin{aligned}
> A_2
> & \le \frac{\eta^2}{m^2} \mathbb{E}\_t\left[\left\|\sum\_{i=1}^m \sum\_{k=0}^{K-1} \mathbf{g}\_{t, k}^i\right\|^2\right] +
> \frac{\eta^2}{m^2}  \mathbb{E}\_t \left[ \left\| \sum_{i=1}^m \sum\_{k=0}^{K-1} n_{t, k}^i \right\|^2\right] \\\\
> & = \frac{\eta^2}{m^2} \mathbb{E}\_t\left[\left\|\sum\_{i=1}^m \sum_{k=0}^{K-1} \mathbf{g}\_{t, k}^i\right\|^2\right] +
> \frac{\eta^2}{m^2} K \cdot \mathbb{E}\_t \left[ \left\| \sum\_{i=1}^m n_t^i \right\|^2\right]
> \end{aligned}
> $$
>
> Considering the quantization noise on each client can be supposed independent, the expectation in the second term is calculated as:
>
> $$
> \begin{aligned}
> \mathbb{E}\_t \left[ \left\| \sum\_{i=1}^m n_t^i \right\|^2\right]  = \sum\_{i=1}^m \mathbb{E}\_t \left[ \left\|  n_t^i \right\|^2\right] + 2\sum_{i<j} \mathbb{E}_t\left[n_t^i n_t^j\right] = \sum\_{i=1}^m ((\mu\_N^i)^2 + (\sigma^i\_N)^2) + 2\sum\_{i<j} \mu_N^i \mu_N^j
> \end{aligned}
> $$
>
> Therefore, $\Phi$ in Equation (10) changes as
>
> $$
> \begin{aligned}
> \Phi =& \frac{1}{c}\left[\frac{L \eta}{2 m} \left(\sigma_L^2 + \frac{1}{m} \left( \sum\_{i=1}^m  ((\mu\_N^i)^2 + (\sigma^i\_N)^2) \right)\right)   +\frac{5 K \eta^2 L^2}{2}\left(\sigma_L^2+6 K \sigma_G^2\right)  \right]
> \end{aligned}
> $$
>
> However, the convergence rate does not change, since the quantization noise expectation is usually at 0.
>
> Note that previous convergence result is a special case of this new result.
>
> > **W2: References Not Discussed.**
>
> **Response:** Thanks for the reminder. We will include FedAvg for the standard FL setting in the final version.
>
> > **W3: Some typos.**
>
> **Response:** We will correct these typos in the final version.
>
> > **Q1: Bit-width change for other models on other datasets.**
>
> **Response:** Yes, there are no significant differences among clients but there is a difference between activations and gradients for other models on other datasets. In the table below, we provide the bit-width change process for training ViT-S on CIFAR-10 (200 epochs) and LeNet on CIFAR-100 (2000 epochs) as further evidence. The numbers in brackets indicate the epochs of bit-width change.
>
> | Model | Dataset | Client Idx | Activation | Gradient |
> | ---- | ---- | ---- | ---- | ---- |
> | ViT-S | CIFAR-10 | 2 | (85, 110) | (20, 85) |
> |  | | 6 | (80, 105) | (20, 80) |
> |  |  | 8 | (80, 105) | (20, 75) |
> | LeNet | CIFAR-100 | 20 | (850, 1345) | (170, 825) |
> | | | 60 | (830, 1325) | (150, 830) |
> | | | 80 | (825, 1355) | (140, 820) |
>
> > **Q2: Regarding quantization error as Gaussian noise.**
>
> **Response:** First of all, we emphasize that we no longer assume Gaussian noise for quantization error to achieve the convergence result. We assume that the expectation and variance of the noise distribution on each client exist are well defined and the expectation can be viewed as zero. We further perform experiments with ResNet-18 on CIFAR-10 to support the assumption, where the distribution of quantization noise is concentrated around zero.
>
> In the previous result, we made the assumption as in [B1]-[B4] where the Gaussian distribution is assumed to find the optimal clipping value to reduce the quantization loss.
>
> We will revise the assumption and clarify claims for notations in the final version.
>
> [B1] Ron Banner et al. Scalable methods for 8-bit training of neural networks. NeurIPS 2018.
>
> [B2] Ruizhou Ding et al. Regularizing activation distribution for training binarized deep networks. CVPR 2019.
>
> [B3] Zhezhi He and Deliang Fan. Simultaneously optimizing weight and quantizer of ternary neural network using truncated gaussian approximation. CVPR 2019.
>
> [B4] Xishan Zhang et al. Fixed-point back-propagation training. CVPR 2020.

---

### Official Review · Reviewer_EVpv · 2025-03-25

**Overall Recommendation:** 3

**Summary:**

This paper introduces LBI-FL, a novel framework for low-bit integerized federated learning with a focus on temporally dynamic bit-width allocation.
The authors provide both theoretical convergence analysis and empirical validation on multiple FL benchmarks showing reductions in communication and computational cosr while maintaining model accuracy.

**Claims And Evidence:**

Most of the claims are well supported.

The selection of the dataset is limited to CIFAR-10 and CIFAR-100, which are both small and lacks evaluation on fundamentally different tasks.

**Essential References Not Discussed:**

Layer-wise quantization based methods are mentioned but not used as baselines.

Some recent works focus on energy-latency tradeoffs in FL with quantization can also be discussed here.

**Experimental Designs Or Analyses:**

The experiment design are valid.

The generalization of RL policy to unseen FL tasks is not fully evaluated which limits the claim of broader usage.

**Methods And Evaluation Criteria:**

The methods and evaluation are appropriate and make sense. The models selected are diversified, and the metrics support the claims.

**Other Comments Or Suggestions:**

More diversified datasets are recommended in this paper

**Other Strengths And Weaknesses:**

Strengths:
The idea of this paper seems novel, well-written with strong theoretical contribution
Weakness:
RL part is not fully discussed.

**Questions For Authors:**

How does the RL agent's decision-making overhead compare to the computational savings from low-bit training?

**Relation To Broader Scientific Literature:**

This work is closely related to FL Basic, theory, quantization and RL for optimization.

This work extends prior works on low-bit FL by proposing a sub-INT8 training framework that dynamically adjusts bit-width over time using RL seems novel to me.

**Theoretical Claims:**

Theoretical contribution of Theorem 4.3, which analyzes convergence of LBI-FL under gradient quantization with Gaussian noise approximation, looks correct.
The convergence rate of $O(1/\sqrt(T))$ makes sense.

---

> ### Author Rebuttal · Authors · 2025-04-01
>
> Thank you for your valuable comments and below are our detailed responses to the raised weaknesses and questions.
>
> >  **W1: Experiments on diverse tasks and datasets.**
>
> **Response:** We have performed extensive experiments on diverse tasks and datasets to demonstrate the effectiveness of our LBI-FL. We use the RL agent trained for image classification on CIFAR-10 without retraining, which proves the generality of our RL agent.
>
> i) Image segmentation with U-Net on DSB2018. We adopt the image size of 96 $\times$ 96. The table below shows that, compared with UI8 training, we achieve acceptable performance of about 0.6\% loss in Dice similarity coefficient (DSC) with over 50\% reduction of BitOPs.
>
> | Dataset | Method | Dice (\%) | BOPS | RR (\%) |
> | ---- | ---- | ---- | ---- | ---- |
> | DSB2018 | FP32 | 89.55 | 19848G | - |
> |  | UI4 | 87.01 | 310.1G | 75 |
> |  | UI8 | 89.48 | 1240.5G | 0 |
> |  | LBI-FL | 88.84 | 604.9G | 51.27 |
>
> ii) Image classification with ResNet-18 on Tiny-ImageNet. The image size is 64$\times$64. Training on Tiny-ImageNet is much more difficult. The table below shows that our LBI-FL achieves over 50\% reduction of BitOPs with less than 0.1\% accuracy loss.
>
> | Dataset | Method | Acc (\%) | BOPS | RR (\%) |
> | ---- | ---- | ---- | ---- | ---- |
> | Tiny-ImageNet | FP32 | 35.84 | 457.14G | - |
> |  | UI4 | 34.64 | 7.14G | 75 |
> |  | UI8 | 35.21 | 28.57G | 0 |
> |  | LBI-FL | 35.13 | 14.05G | 50.81 |
>
> > **W2: Baselines using layer-wise quantization.**
>
> **Response:** Most layer-wise quantization based methods (**e.g.**, HAWQ v1-v3) are designed for inference only, and cannot be employed for FL. We compare our LBI-FL with AMPA [A1] that achieves layer-wise quantization based on the sensitivity measurement for centralized low-bit training. We train ResNet-18, MobileNet-V2, and ViT-S on CIFAR-10 using the two methods. The results below show that our LBI-FL achieves higher compression ratios and better performance.
>
> [A1] Li Ding et al. AMPA: Adaptive mixed precision allocation for low-bit integer training. ICML 2024.
>
> | Model | Method | Acc (\%) | BOPS | RR (\%) |
> | ---- | ---- | ---- | ---- | ---- |
> | ResNet-18 | AMPA | 84.10 | 3.49G | 51.11 |
> | | LBI-FL | 84.16 | 3.28G | 54.06 |
> | ViT-S | AMPA | 72.54 | 73.08G | 38.79 |
> |  | LBI-FL | 72.55 | 60.3G | 49.51 |
> | MobileNet-V2 | AMPA | 87.89 | 9.78G | 46.19 |
> |  | LBI-FL | 89.02 | 8.83G | 51.39 |
>
> > **W3: Discussion about works on energy-latency tradeoffs with quantization in FL.**
>
> **Response:** Thanks for the suggestion. We will include relevant works [A2][A3] in the Related Work section. Optimizing energy consumption is indeed essential for practical deployment of FL systems. For instance, [A2] derives the time and energy consumption models for FL and proposes a iterative algorithm to allocate resources. [A3] proposes an optimization framework that minimizes the total energy consumption of local computation and wireless transmission by adaptively selecting the quantization level. These methods are complementary to our LBI-FL that simultaneously reduces computational and communication overheads, and thereby improves energy efficiency with reduced latency in federated learning.
>
> [A2] Yang, Zhaohui, et al. "Energy efficient federated learning over wireless communication networks." IEEE TWC 2020.
>
> [A3] Ouiame Marnissi, Hajar El Hammouti, and El Houcine Bergou. Adaptive sparsification and quantization for enhanced energy efficiency in federated learning.IEEE OJCOMS 2024.
>
> > **Q1: Overhead of RL agent's decision-making.**
>
> **Response:** The overhead of RL agent is very small compared to the computational savings from low-bit training.
>
> i) The RL agent consists of two linear layers with only 1.92K parameters, and requires 1.97G BitOPs for making one decision (i.e., 1.92K Mac$\times$32$\times$32).
>
> ii) The RL agent makes decision for every 5 epochs rather than each epoch.
>
> We provide results on CIFAR-10 as examples. When training LeNet using 100 clients, each client has 500 training images and the RL agent makes a decision after training on 250 images on average (participation rate is 0.1). Its inference cost is only 1.1\% that for UI4 training and 0.275\% for UI8 training. When training ResNet-18 using 10 clients, each client has 5000 training images. The inference cost of the RL agent is 0.0044\% and 0.0011\% those for UI4 and UI8 training.

---

### Decision · Program_Chairs · 2025-05-01

**Decision:**

Accept (poster)

**Comment:**

The reviewers identified the novelty of the proposed framework and its contributions to communication-efficient federated learning, and were consistent to accept this paper. It would be beneficial if the responses addressing the reviewers’ questions are added into the final version.